- Characteristics and processing of aqueous secondary organic aerosols
- during autumn in suburban Eastern China: role of aerosol liquid
- water, aerosol acidity, and photochemistry
- Qiu Wang<sup>1</sup>, Tengyu Liu<sup>1,2,3\*</sup>, Weiqi Xu<sup>4</sup>, Jinbo Wang<sup>1,5</sup>, Dafeng Ge<sup>1</sup>, Caijun Zhu<sup>1</sup>,
- Chuanhua Ren<sup>1</sup>, Jiaping Wang<sup>1,2</sup>, Qiaozhi Zha<sup>1,2</sup>, Ximeng Qi<sup>1,2</sup>, Wei Nie<sup>1,2</sup>, Xuguang
- Chi<sup>1,2</sup>, Sijia Lou<sup>1,2,3</sup>, Xin Huang<sup>1,2,3</sup>, Aijun Ding<sup>1,2,3</sup>
- <sup>1</sup>School of Atmospheric Sciences, Nanjing University, Nanjing, 210023, China
- <sup>2</sup>National Observation and Research Station for Atmospheric Processes and
- Environmental Change in Yangtze River Delta, Nanjing, 210023, China
- <sup>3</sup>Frontiers Science Center for Critical Earth Material Cycling, Nanjing University,
- Nanjing, 210023, China
- <sup>4</sup>State Key Laboratory of Atmospheric Environment and Extreme Meteorology,
- Institute of Atmospheric Physics, Chinese Academy of Sciences, Beijing 100029, China
- 5Now at National Satellite Meteorological Center, China Meteorological
- Administration, Beijing 100081, China
- \*Corresponding author: Tengyu Liu (tengyu.liu@nju.edu.cn)

#### Abstract

Aqueous-phase secondary organic aerosols (aqSOA) constitute a large fraction of SOA, thereby exerting significant influence on air quality, climate, and human health. However, its formation mechanisms remain unclear due to limited observational evidence. We conducted field measurements of particulate matter (PM) composition by deploying high-resolution aerosol mass spectrometry in a suburban environment during autumn in Nanjing, China. The characteristics and formation pathways of aqSOA are comprehensively investigated by using Positive Matrix Factorization (PMF) method. Our results show that aqSOA accounted for 27.6% of oxidized organic aerosols, exhibiting elevated O:C ratios (0.78) and strong correlations with nitrate and aerosol liquid water (ALW). The important role of acid-catalyzed reactions is also revealed by the enhanced production of aqSOA at lower aerosol pH conditions. Under elevated nitrate and ALW levels, a pronounced morning agSOA peak was frequently observed; whereas a noon-time aqSOA peak was also observed on several days, likely governed by photochemistry and aqueous-phase reactions. These findings highlight the critical roles of nitrate, ALW, acidity, and photochemistry in driving aqSOA production in polluted urban environments. This study advances the mechanistic understanding of aqSOA formation and provides insights into the mitigation of SOA in Eastern China.

### 1 Introduction

36

Organic aerosols (OA) significantly affect air quality, human health, and climate by 38 influencing radiative forcing and cloud formation (Kanakidou et al., 2005; Zhou et al., 39 2019). Secondary organic aerosols (SOA) can contribute 30-70% of total organic 40 aerosols (Huang et al., 2014; Xian et al., 2023). The formation of SOA has been 41 attributed mainly to gas-phase oxidation of volatile organic compounds (VOCs), where 42 the oxidized products subsequently partition into the aerosol phase (Hennigan et al., 43 2009; Seinfeld and Pankow, 2003; Ziemann and Atkinson, 2012). Recently, growing 44 evidence highlights the important role of SOA generated through aqueous-phase 45 processes in cloud droplets, fog, and aerosol liquid water (ALW) (Ervens et al., 2011; 46 Kim et al., 2019; Sun et al., 2010). However, the formation mechanisms and sources of 47 aqueous-phase SOA (aqSOA) remain highly uncertain (Ervens et al., 2011; Huang et 48 al., 2025; McNeill, 2015). 49 In recent years, concentrations of OA have gradually declined across many regions 50 in China due to the implementation of air pollution control measures, but the relative 51 contribution of SOA has increased markedly (Chen et al., 2024), with aqSOA 52 constituting a substantial fraction. Numerous studies have demonstrated that elevated 53 ALW can significantly promote aqSOA formation during winter haze episodes through 54 multiphase reactions (Chen et al., 2021; Feng et al., 2022; Liu et al., 2019; Peng et al., 55 2021; Sun et al., 2016, 2019; Wang et al., 2023; Xiao et al., 2022; Xu et al., 2017; Zhao 56 et al., 2019). Still, the exact processes driving aqSOA formation is not fully 57 characterized. Field observations in urban Beijing have shown that ring-breaking

oxidation and functionalization of polycyclic aromatic hydrocarbons of fossil-fuel-59 derived primary organic aerosols could lead to rapid agSOA formation at high relative 60 humidity (RH) during winter haze episodes (Wang et al., 2021). Another field study 61 conducted in the North China Plain showed that the formation of aqSOA could be 62 largely enhanced under favorable photochemical conditions with precursors originated 63 from biomass burning activities (Kuang et al., 2020). Recently, field measurements in 64 Hebei reported that high nitrate may support the potential formation/transformation 65 from POA-related components to aqSOA (Gu et al., 2023). Laboratory studies reveal 66 that accretion reactions, which play a crucial role in SOA formation, are highly sensitive 67 to pH levels (Tilgner et al., 2021). Moreover, under the emerging dominance of nitrate 68 in aerosol composition (Huang et al., 2025), the interplay among nitrate, ALW, and pH 69 may complex the agSOA formation and requires further investigation. 70 The Yangtze River Delta (YRD) region is one of the most densely populated and 71 economically developed areas in China, characterized by intensive industrial activity, 72 heavy traffic emissions, frequent regional pollution episodes, and high relative 73 humidity (Liu et al., 2025). Several previous studies have examined aqSOA processes 74 in this region (Wang et al., 2016; Wu et al., 2018; Xian et al., 2023), but uncertainties 75 regarding its sources, controlling factors, and formation mechanisms still remained. 76 Consequently, it is challenging for current models to precisely simulate aqSOA and its 77 contribution to the YRD region (Ervens et al., 2011; Rogers et al., 2025). 78 In this study, we conducted real-time measurements of OA at the National 79 Observation and Research Station for Atmospheric Processes and Environmental

80 Change in Yangtze River Delta (SORPES) located in suburban Nanjing in the western 81 YRD region during the autumn of 2020. The chemical characteristics and formation 82 mechanisms of agSOA, as well as the roles of nitrate, ALW, and aerosol acidity are 83 investigated. By classifying diurnal variation patterns of aqSOA, we assessed the 84 relative roles of photochemical and aqueous-phase processes. The results provide new 85 insights into regional aqSOA formation in the YRD and have implications for the 86 development of effective air pollution control strategies. 87 2 Materials and Methods 88 2.1 Sampling Site 89 The field campaign was conducted from 13 October to 30 December in 2020 at 90 SORPES station (118°57'E, 32°07'N) located in the Xianlin campus of Nanjing 91 University in Nanjing, China. This is a representative station of western YRD, 92 surrounded by high vegetation cover, and also subject to more anthropogenic emissions 93 (Ding et al., 2016, 2019; Dou et al., 2025; Liu et al., 2025). 94 2.2 Instrumentation 95 Real-time non-refractory PM1 composition was measured using a high-resolution time-96 of-flight aerosol mass spectrometer (HR-ToF-AMS; hereafter, AMS; Aerodyne 97 Research Inc.). An aerodynamic PM<sub>1</sub> lens was used to focus the particle into a beam, 98 which was then impacted on the heated tungsten surface (~600°C) and flash-vaporized. 99 In our study, ambient aerosols were passed through a ~2 m long stainless-steel sampling 100 tube, dried by a Nafion drying tube, and then introduced into the AMS. In order to 101 obtain highly sensitive data, AMS was operated in V mode with a time resolution of 2

102 minutes (DeCarlo et al., 2006). 103 Other instruments were also employed at SORPES in support of these 104 measurements. Black carbon (BC) was measured by the photoacoustic extinctiometer 105 (PAX, Droplet Measurement Technologies Inc., USA). The meteorological parameters 106 and gaseous pollutants were also measured simultaneously. Ozone (O3), carbon 107 monoxide (CO), nitric oxide (NO), nitrogen oxides (NO<sub>x</sub>) and sulfur dioxide (SO<sub>2</sub>) 108 were measured using online analyzers (Thermo Fisher Scientific, USA). Ammonia 109 (NH<sub>3</sub>) was measured by the Picarro G2103 gas analyzer (Picarro Inc., USA) (Liu et al., 110 2024). Temperature, RH and other meteorological parameters were monitored by 111 meteorological sensors (GRWS100, Campbell, USA). 112 2.3 Data Analysis 113 The AMS data were processed by SQUIRREL (version 1.60P) and PIKA (version 1.20P) 114 from the ToF-AMS Software Downloads Web page (http://cires.colorado.edu/jimenez-115 group/ToFAMSResources/ToFSoftware/index.html). The ionization efficiency (IE) 116 was calibrated using 300 nm pure ammonium nitrate before and after the campaign. 117 The relative ionization efficiency (RIE) of ammonium was determined from pure 118 ammonium nitrate, yielding a value of 3.52. RIE values for OA, nitrate, sulfate, and 119 chloride were set to their default values of 1.4, 1.1, 1.2, and 1.3, respectively 120 (Canagaratna et al., 2007). As well, the collection efficiency (CE) was assigned a typical 121 value of 0.5 for common environments. Element ratios, including H: C, O: C, N: C and 122 OM: OC, are calculated using the Improved-Ambient method (Canagaratna et al., 2015). 123 In addition, ALW content and aerosol acidity that are associated with inorganic

124 species were estimated by the Extended Aerosol Inorganics Model (E-AIM), which is 125 a well-known inorganic thermodynamic model without simplifying assumptions (Clegg 126 et al. 1998; Wexler and Clegg 2002; Pye et al. 2020). 127 2.4 Source Apportionment of OA 128 Positive Matrix Factorization (PMF) analysis was applied to the high-resolution mass 129 spectra of organic matrix for m/z 12 - 120 to resolve distinct OA factors from specific 130 sources (Paatero and Tapper, 1994; Ulbrich et al., 2009). The data and error matrices 131 were treated according to the procedures detailed in DeCarlo et al. (2010). By 132 comparing the mass spectral profiles with previous studies and correlations with time 133 series of tracers, five OA factors with fpeak = 0 were selected, including one primary 134 organic aerosol (POA) factor, one nitrogenous OA (NOA) factor, and three SOA factors, 135 namely, less-oxidized oxygenated OA (LO-OOA), more-oxidized OOA (MO-OOA), 136 and aqSOA. The detailed diagnostic plots are shown in Figures S1 and S2 in the 137 supporting information. 138 Among these five OA factors, aqSOA exhibits typical characteristics of highly 139 oxidized organic aerosols: fraction of m/z 44 (CO<sub>2</sub>+, primarily from carboxylic acids 140 and highly oxidized compounds) (Heald et al., 2010) in total signals exceeded that of 141 m/z 43 (typically representing less oxidized compounds) (Figure S2). Elemental 142 analysis further indicated a strongly oxygenated character, with an average H:C ratio of 143 1.80 and an O:C ratio of 0.78. These values are comparable to those reported in other 144 regions, such as northern Italy and Beijing (Gilardoni et al., 2016; Xu et al., 2019). The 145 N:C ratio (0.05) was also elevated and similar to wintertime values observed in Beijing

146 (N:C = 0.045; Xu et al. 2017), suggesting nitrogen-containing compound formation via 147 aqueous-phase processing. AqSOA exhibited strong correlations with unique fragment 148 ions that are widely recognized as markers of aqueous-phase secondary products (Sun 149 et al., 2016; Xu et al., 2019). For instance, significant correlations were observed with 150  $C_2O_2^+$  (m/z 56, r = 0.77), a typical fragment of oxalate-related species, and with 151  $CH_3SO^+$  (m/z 63, r = 0.90), which is indicative of organosulfur compounds (Figure S3). 152 These results provide chemical evidence supporting the aqueous-phase origin of aqSOA 153 in Nanjing. 154 3. Results and Discussions 155 3.1 General Characteristics of agSOA 156 The meteorological conditions in Nanjing were overall stable during the field campaign 157 (October-December), with an average temperature and RH of  $14.6 \pm 4.6$  °C and  $65.1 \pm$ 158 17.9%, respectively (Figure S4). The diurnal variation of RH ranged from 50% to 85%, 159 resulting in relatively humid air conditions that favored aqueous-phase chemical 160 reactions. The average wind speed of the prevailing northerly wind was relatively low 161  $(0.21 \pm 0.16 \,\mathrm{m/s})$ , resulting in the accumulation of local pollutants. In particular, the 162 average NR-PM<sub>1</sub> concentration was  $37.3 \pm 20.6 \,\mu\text{g/m}^3$ , indicating frequently occurred 163 particulate matter pollution during this period. 164 Analysis of aerosol chemical composition revealed that organic aerosol was the 165 dominant NR-PM<sub>1</sub> component in Nanjing, accounting for 40.9% of the total PM<sub>1</sub> 166 concentration, evidently higher than nitrate (30.7%) and sulfate (13.9%; Figure S4). 167 The time series of aqSOA showed significant variation, with peak concentrations up to

15.7 μg/m³ (Figure 1a). The average concentration of aqSOA during the campaign was 3.1 µg/m<sup>3</sup>, accounting for 20.2% (62.7% in maximum) of the total OA and 27.6% (78.2% in maximum) of the total SOA (Figure 1b). The average fraction of aqSOA in OA was much higher than Beijing (13-17%) (Zhao et al., 2019), indicating an important role of aqueous-phase processes in SOA formation in Nanjing. Moreover, both aqSOA concentrations and their relative contribution to total organic aerosol increased significantly with increasing RH (Figure S5a), which was likely due to the increased availability of the aqueous reaction medium enhanced by water uptake by aerosols, which is crucial for the aqSOA formation. This is further confirmed with the sharply increased contribution of agSOA to OA (up to 62.7%) at RH levels above 80%, suggesting that high-humidity environments in suburban Nanjing substantially promoted aqSOA formation through enhanced aqueous-phase chemistry. The Van Krevelen diagram (Heald et al., 2010) provided further insights into the chemical aging of OA (Figure S6). The H:C vs. O:C slope was near -0.5, indicating OA oxidation primarily involving carboxylic acid and peroxide or alcohol functional groups addition without fragmentation and/or the addition of carboxylic acid functional groups with fragmentation (Ng et al., 2011). In particular, OA observed under high RH conditions (>80%) clustered in the upper plot region, suggesting distinct chemical evolution of OA when aqueous-phase reactions were involved. Specifically, the nearly zero slope of the relationship among POA, LO-OOA and aqSOA suggests that the observed increase in the O:C ratio of aqSOA may be related to oligomerization and hydroxyl formation through dark chemistry processes (Lim et al., 2010).

**Figure 1.** (a) Time series of aqSOA and nitrate concentrations. The pink area represents diurnal pattern type I, while the grey area represents diurnal pattern type II. (b) Time series of the fraction of five OA factors.

## 3.2 Enhanced aqSOA Formation Driven by Nitrate, ALW, and Acid

## Catalysis

As a strongly hygroscopic component, nitrate aerosol can enhance aerosol water uptake, thereby modifying the aqueous microenvironment for SOA production (Hodas et al., 2014; Sullivan et al., 2016). We observed strong correlation between aqSOA and nitrate concentrations (R<sup>2</sup> = 0.83; Figure 2a) during the observation period, and the variation of aqSOA was highly consistent with that of nitrate aerosol (Figure 1a). This indicates that, nitrate aerosols play an important role in the aqSOA formation in Nanjing, likely through influencing ALW and/or aqueous reactions.

**Figure 2**. (a) Scatter plot of aqSOA and nitrate concentrations. (b) Scatter plot of aqSOA and ALW concentrations, colored by aerosol pH. The blue line represents the fitted line for data with aerosol pH < 3, while the red line represents the fitted line for data with aerosol pH > 3.

The correlation between aqSOA and aerosol liquid water (ALW; r = 0.63; Figure 2b), which is mainly derived from the hygroscopic growth of inorganic salts such as nitrate, was moderate compared to nitrate. Still, their positive relationship indicates that higher ALW levels may enhance aqueous-phase chemical processes by providing the medium in which multiphase reactions can occur (Hodas et al., 2014). The presence of ALW promotes the partitioning of water-soluble organic precursors and supports subsequent aqueous-phase reactions leading to aqSOA formation. The results are consistent with previous studies in humid urban environments (Chen et al., 2021; Duan et al., 2022; Kuang et al., 2020).

In addition to ALW, aerosol acidity also exerted a strong influence on aqSOA yields during the campaign (Lim et al., 2010; Tilgner et al., 2021). To illustrate, the

dataset was separated by aerosol pH (pH < 3 vs. pH > 3) (Figure 2b), the slopes of ALW-aqSOA correlations decreased with increasing pH, indicating that aqSOA production was higher under more acidic conditions at the same ALW level. This pattern demonstrates the importance of acid-catalyzed reactions in driving aqSOA formation. Previous studies have demonstrated that under acidic conditions (low pH, high H+ concentration), non-oxidative aqueous organic chemical processes, such as accretion reactions (aldol condensation, hemiacetal and acetal formation, and the esterification of carboxylic acids) are important formation pathways of aqSOA (Freedman et al., 2019; Tilgner et al., 2021). For example, the hydration of methylglyoxal and its subsequent acetal formation are highly pH-dependent, requiring a pH of less than 3.5 to occur (Yasmeen et al., 2010). As aerosol pH increases (lower H<sup>+</sup> availability), the catalytic efficiency of these acid-driven reactions diminishes, leading to a reduction in aqSOA production efficiency Collectively, these findings suggest that aerosol acidity plays a pivotal regulatory role in aqSOA formation, linking aqueous chemistry to the broader context of aerosol physicochemical properties in Nanjing. Thus, our measurements revealed a synergistic interplay among nitrate, ALW, and aerosol acidity in regulating aqSOA formation in Nanjing. Nitrate enhances the ALW content, which in turn promotes aqueous-phase reactions, while aerosol acidity governs the efficiency of these chemical processes. These processes together constituted the formation mechanism of agSOA during the observation period.

3.3 Synergistic Role of Aqueous and Photochemical Processes 245 During the campaign, the average diurnal variation of aqSOA in Nanjing exhibited a 246 distinct morning peak at approximately 09:00 local time (Figure S2d). This pattern is 247 different from most of the previous urban observations, where aqSOA concentrations 248 typically peak during nighttime periods (Sun et al., 2016; Xu et al., 2019; Gu et al., 249 2023). To address this, the time series of aqSOA was analyzed on a daily basis, with 250 rainy days excluded. Based on this analysis, two distinct diurnal variation patterns were 251 identified: type I (22 days) and type II (9 days) (Figure 1a). For diurnal pattern type I, 252 the daily variation of aqSOA exhibits a pronounced morning peak, whereas the daytime 253 concentration of aqSOA shows a noon peak for diurnal pattern type II (Figure 3). These 254 two distinct diurnal patterns of aqSOA could be attributed to multiple formation 255 mechanisms of agSOA or different meteorological conditions. 256 During diurnal pattern type I, agSOA concentrations exhibited a pronounced 257 single peak between 09:00 and 10:00 local time, following a period of continuous 258 nighttime accumulation from approximately 20:00 to 07:00 (Figure 3a). Notably, the 259 aqSOA peak lagged behind the nitrate peak by about one hour, and nocturnal increases 260 in nitrate, ALW, and aqSOA were highly synchronized. Compared with type II, 261 nighttime meteorological conditions during type I were characterized by higher RH 262 (73.1% vs. 61.2%) and ALW concentrations (34.2  $\mu$ g/m³ vs. 12.6  $\mu$ g/m³). Such humid 263 conditions favor the formation of both nitrate and ALW, thereby enhancing aqSOA 264 production through aqueous-phase reactions. Moreover, the higher RH conditions 265 observed during type I events could further promote aqueous reaction pathways, as the

nitrate formation from  $N_2O_5$  hydrolysis on aqueous aerosol surfaces is strongly facilitated under such conditions (Sun et al., 2018). Overall, these observations indicate that aqSOA production under type I conditions is predominantly controlled by aerosol aqueous-phase processes, highlighting the important role of nocturnal multiphase chemistry in driving morning aqSOA peaks.

Figure 3. Diurnal variations for SOA, ALW, and nitrate in diurnal pattern types I (a) and II (b). Diurnal variations for temperature, humidity, and  $O_x$  concentration in diurnal pattern types I (c) and II (d).

In contrast, aqSOA concentrations gradually increased during the daytime and peaked around 13:00 local time (Figure 3b) during diurnal pattern type II, indicating an

influence of photochemical processes. The diurnal variation of aqSOA is consistent with that of nitrate aerosols. This suggest that photochemistry leads to the formation of nitrate aerosols and likely further mediate the aqSOA formation through water uptake. Ox is commonly regarded as a tracer for photochemical processes. The relationship between daytime (08:00–17:00) aqSOA concentrations and Ox levels was examined for both diurnal patterns (Figure 4). For diurnal type I, no significant correlation was observed, suggesting that daytime photochemistry played a minimal role in aqSOA production under these conditions. In contrast, diurnal type II displayed a positive correlation, implying that aqSOA formation was partially related to the photochemical processes.

Figure 4. Scatter plots of aqSOA and  $O_x$  for Diurnal pattern types I (a) and II (b). Data points are colored by RH, with red and blue lines representing fits for RH > 60% and RH 

Further analysis revealed that the slope of the aqSOA-O<sub>x</sub> relationship under high RH conditions in diurnal type II was approximately three times higher than that under low RH conditions. This is because higher RH likely enhanced the partitioning of semi-volatile photochemical oxidation products into the aqueous phase and promoted aqueous-phase photochemical reactions, thereby facilitating aqSOA formation. These results indicate that daytime aqSOA production during type II is governed by the combined effects of photochemical oxidation and humidity-dependent aqueous processing.

#### 4. Conclusions

We analyzed the characteristics and formation mechanisms of aqSOA in the autumn season in Nanjing, where aqSOA was found to constitute a significant fraction of organic aerosols (20.2%). The results demonstrate that nitrate, ALW, and aerosol acidity act in concert to regulate aqSOA formation, with nitrate serving as a critical driver through its strong hygroscopicity. We observed the occurrence of a distinct morning peak of aqSOA in the YRD region, a feature that differs from most of the previously reported studies in urban environments. Based on diurnal variation analysis, two distinct diurnal patterns were identified: type I, dominated by nighttime aqueousphase chemistry linked to nitrate and ALW accumulation; and Type II, shaped primarily by daytime photochemical oxidation. These findings highlight that aqSOA formation in this megacity is governed by the synergistic effects of both nocturnal aqueous reactions and daytime photochemical processes.

317 China is strictly controlling SO<sub>2</sub> emissions, which has led to a decrease in sulfate 318 content in atmospheric particulate matter. Nitrate is becoming a more important 319 inorganic component of PM<sub>2.5</sub>, and its role in promoting aqSOA formation will become 320 even more significant. Although this study did not directly resolve the specific 321 precursor compounds of aqSOA, our results still highlight the importance of paying 322 greater attention to nitrate-driven aqueous processes in future air quality assessments. 323 In particular, coordinated management of nitrogen oxides (NO<sub>x</sub>) alongside traditional 324 particulate matter controls may provide an effective pathway for mitigating aqSOA 325 burdens in a post-sulfate-dominated atmosphere. 326 327 Data availability 328 All data are available from the corresponding author upon request. 329 **Author contributions** 330 T.L. and A.D. designed the research project; J.W., D.G., C.Z., C.R., W.X., J.W., W.N., 331 X.C., S.L. and X.H. performed the research; Q.W., Q.Z., and X.Q. analyzed data; Q.W., 332 Q.Z., and T.L. wrote the paper. All authors participated in the relevant scientific 333 discussion and commented on the manuscript. 334 **Competing interests** 335 The authors declare no competing interests. 336 Acknowledgments 337 The authors thank the National Natural Science Foundation of China project (42222504) 338 and the Fundamental Research Funds for the Central Universities (14380221, 339 14380237) for funding.

References 341 Canagaratna, M. r., Jayne, J. t., Jimenez, J. l., Allan, J. d., Alfarra, M. r., Zhang, Q., 342 Onasch, T. b., Drewnick, F., Coe, H., Middlebrook, A., Delia, A., Williams, L. r., 343 Trimborn, A. m., Northway, M. j., DeCarlo, P. f., Kolb, C. e., Davidovits, P., and 344 Worsnop, D. r.: Chemical and microphysical characterization of ambient aerosols with 345 the aerodyne aerosol mass spectrometer, Mass Spectrometry Reviews, 26, 185–222, 346 https://doi.org/10.1002/mas.20115, 2007. 347 Canagaratna, M. R., Jimenez, J. L., Kroll, J. H., Chen, Q., Kessler, S. H., Massoli, P., 348 Hildebrandt Ruiz, L., Fortner, E., Williams, L. R., Wilson, K. R., Surratt, J. D., Donahue, 349 N. M., Jayne, J. T., and Worsnop, D. R.: Elemental ratio measurements of organic 350 compounds using aerosol mass spectrometry: Characterization, improved calibration, 351 253-272, implications, Atmospheric and Chemistry and Physics, 15, 352 https://doi.org/10.5194/acp-15-253-2015, 2015. 353 Chen, C., Zhang, H., Yan, W., Wu, N., Zhang, Q., and He, K.: Aerosol water content 354 enhancement leads to changes in the major formation mechanisms of nitrate and 355 secondary organic aerosols in winter over the North China Plain, Environmental 356 Pollution, 287, 117625, https://doi.org/10.1016/j.envpol.2021.117625, 2021. 357 Chen, Q., Miao, R., Geng, G., Shrivastava, M., Dao, X., Xu, B., Sun, J., Zhang, X., Liu, 358 M., Tang, G., Tang, Q., Hu, H., Huang, R.-J., Wang, H., Zheng, Y., Qin, Y., Guo, S., Hu, 359 M., and Zhu, T.: Widespread 2013-2020 decreases and reduction challenges of organic 360 aerosol in China, Nat Commun, 15, 4465, https://doi.org/10.1038/s41467-024-48902-361 0, 2024.

- Clegg, S. L., Brimblecombe, P., and Wexler, A. S.: Thermodynamic model of the system
- H<sup>+</sup>-NH<sub>4</sub><sup>+</sup>-SO<sub>4</sub><sup>2</sup>-NO<sub>3</sub><sup>-</sup>-H<sub>2</sub>O at tropospheric temperatures, J. Phys. Chem. A, 102,
- 2137–2154, https://doi.org/10.1021/jp973042r, 1998.
- DeCarlo, P. F., Kimmel, J. R., Trimborn, A., Northway, M. J., Jayne, J. T., Aiken, A. C.,
- Gonin, M., Fuhrer, K., Horvath, T., Docherty, K. S., Worsnop, D. R., and Jimenez, J. L.:
- Field-deployable, high-resolution, time-of-flight aerosol mass spectrometer, Anal.
- Chem., 78, 8281–8289, https://doi.org/10.1021/ac061249n, 2006.
- DeCarlo, P. F., Ulbrich, I. M., Crounse, J., de Foy, B., Dunlea, E. J., Aiken, A. C., Knapp,
- D., Weinheimer, A. J., Campos, T., Wennberg, P. O., and Jimenez, J. L.: Investigation
- of the sources and processing of organic aerosol over the central mexican plateau from
- aircraft measurements during MILAGRO, Atmospheric Chemistry and Physics, 10,
- 5257–5280, https://doi.org/10.5194/acp-10-5257-2010, 2010.
- Ding, A., Nie, W., Huang, X., Chi, X., Sun, J., Kerminen, V.-M., Xu, Z., Guo, W., Petäjä,
- 375 T., Yang, X., Kulmala, M., and Fu, C.: Long-term observation of air pollution-
- weather/climate interactions at the SORPES station: a review and outlook, Front.
- Environ. Sci. Eng., 10, 15, https://doi.org/10.1007/s11783-016-0877-3, 2016.
- Ding, A., Huang, X., Nie, W., Chi, X., Xu, Z., Zheng, L., Xu, Z., Xie, Y., Qi, X., Shen,
- Y., Sun, P., Wang, J., Wang, L., Sun, J., Yang, X.-Q., Qin, W., Zhang, X., Cheng, W.,
- Liu, W., Pan, L., and Fu, C.: Significant reduction of PM<sub>2.5</sub> in eastern China due to
- regional-scale emission control: Evidence from SORPES in 2011-2018, Atmospheric
- Chemistry and Physics, 19, 11791–11801, https://doi.org/10.5194/acp-19-11791-2019,
- 2019.

- Dou, J., Liu, T., Ge, D., Zhang, Y., Yin, J., Wang, L., Liu, H., Li, D., Niu, G., Chen, L.,
- Wang, J., Qi, X., Nie, W., Chi, X., Huang, X., and Ding, A.: In-situ secondary organic
- aerosol formation from ambient air in suburban eastern China: Substantially distinct
- characteristics between summer and winter, Atmospheric Environment, 356, 121295,
- https://doi.org/10.1016/j.atmosenv.2025.121295, 2025.
- Duan, J., Huang, R.-J., Gu, Y., Lin, C., Zhong, H., Xu, W., Liu, Q., You, Y., Ovadnevaite,
- 390 J., Ceburnis, D., Hoffmann, T., and O'Dowd, C.: Measurement report: Large
- contribution of biomass burning and aqueous-phase processes to the wintertime
- secondary organic aerosol formation in Xi'an, Northwest China, Atmos. Chem. Phys.,
- 2022.
- Ervens, B., Turpin, B. J., and Weber, R. J.: Secondary organic aerosol formation in
- cloud droplets and aqueous particles (aqSOA): A review of laboratory, field and model
- studies, Atmospheric Chemistry and Physics, 11, 11069–11102,
- https://doi.org/10.5194/acp-11-11069-2011, 2011.
- Feng, Z., Liu, Y., Zheng, F., Yan, C., Fu, P., Zhang, Y., Lian, C., Wang, W., Cai, J., Du,
- W., Chu, B., Wang, Y., Kangasluoma, J., Bianchi, F., Petäjä, T., and Kulmala, M.:
- Highly oxidized organic aerosols in Beijing: Possible contribution of aqueous-phase
- chemistry, Atmospheric Environment, 273, 118971,
- https://doi.org/10.1016/j.atmosenv.2022.118971, 2022.
- Freedman, M. A., Ott, E.-J. E., and Marak, K. E.: Role of pH in Aerosol Processes and
- Measurement Challenges, J. Phys. Chem. A, 123, 1275–1284,
- https://doi.org/10.1021/acs.jpca.8b10676, 2019.

- Gilardoni, S., Massoli, P., Paglione, M., Giulianelli, L., Carbone, C., Rinaldi, M.,
- Decesari, S., Sandrini, S., Costabile, F., Gobbi, G. P., Pietrogrande, M. C., Visentin, M.,
- Scotto, F., Fuzzi, S., and Facchini, M. C.: Direct observation of aqueous secondary
- organic aerosol from biomass-burning emissions, Proc. Natl. Acad. Sci. U.S.A., 113,
- 10013-10018, https://doi.org/10.1073/pnas.1602212113, 2016.
- Gu, Y., Huang, R.-J., Duan, J., Xu, W., Lin, C., Zhong, H., Wang, Y., Ni, H., Liu, Q.,
- Xu, R., Wang, L., and Li, Y. J.: Multiple pathways for the formation of secondary
- organic aerosol in the North China Plain in summer, Atmospheric Chemistry and
- Physics, 23, 5419–5433, https://doi.org/10.5194/acp-23-5419-2023, 2023.
- Heald, C. L., Kroll, J. H., Jimenez, J. L., Docherty, K. S., DeCarlo, P. F., Aiken, A. C.,
- Chen, Q., Martin, S. T., Farmer, D. K., and Artaxo, P.: A simplified description of the
- evolution of organic aerosol composition in the atmosphere, Geophysical Research
- Letters, 37, https://doi.org/10.1029/2010GL042737, 2010.
- Hennigan, C. J., Bergin, M. H., Russell, A. G., Nenes, A., and Weber, R. J.: Gas/particle
- partitioning of water-soluble organic aerosol in atlanta, Atmospheric Chemistry and
- Physics, 9, 3613–3628, https://doi.org/10.5194/acp-9-3613-2009, 2009.
- Hodas, N., Sullivan, A. P., Skog, K., Keutsch, F. N., Decesari, S., Facchini, M. C.,
- Carlton, A. G., Laaksonen, A., and Turpin, B. J.: Aerosol Liquid Water Driven by
- Anthropogenic Nitrate: Implications for Lifetimes of Water-Soluble Organic Gases and
- Potential for Secondary Organic Aerosol Formation, Environ. Sci. Technol., 2014.
- Huang, R.-J., Zhang, Y., Bozzetti, C., Ho, K.-F., Cao, J.-J., Han, Y., Daellenbach, K. R.,
- Slowik, J. G., Platt, S. M., Canonaco, F., Zotter, P., Wolf, R., Pieber, S. M., Bruns, E.

- 428 A., Crippa, M., Ciarelli, G., Piazzalunga, A., Schwikowski, M., Abbaszade, G.,
- Schnelle-Kreis, J., Zimmermann, R., An, Z., Szidat, S., Baltensperger, U., Haddad, I.
- E., and Prévôt, A. S. H.: High secondary aerosol contribution to particulate pollution
- during haze events in China, Nature, 514, 218–222,
- https://doi.org/10.1038/nature13774, 2014.
- Huang, R.-J., Li, Y. J., Chen, Q., Zhang, Y., Lin, C., Chan, C. K., Yu, J. Z., de Gouw, J.,
- Tong, S., Jiang, J., Wang, W., Ding, X., Wang, X., Ge, M., Zhou, W., Worsnop, D., Boy,
- 435 M., Bilde, M., Dusek, U., Carlton, A. G., Hoffmann, T., McNeill, V. F., and Glasius, M.:
- Secondary organic aerosol in urban China: A distinct chemical regime for air pollution
- studies, Science, 389, eadq2840, https://doi.org/10.1126/science.adq2840, 2025.
- Kanakidou, M., Seinfeld, J. H., Pandis, S. N., Barnes, I., Dentener, F. J., Facchini, M.
- C., Van Dingenen, R., Ervens, B., Nenes, A., Nielsen, C. J., Swietlicki, E., Putaud, J. P.,
- Balkanski, Y., Fuzzi, S., Horth, J., Moortgat, G. K., Winterhalter, R., Myhre, C. E. L.,
- Tsigaridis, K., Vignati, E., Stephanou, E. G., and Wilson, J.: Organic aerosol and global
- climate modelling: A review, Atmospheric Chemistry and Physics, 5, 1053–1123,
- https://doi.org/10.5194/acp-5-1053-2005, 2005.
- Kim, H., Collier, S., Ge, X., Xu, J., Sun, Y., Jiang, W., Wang, Y., Herckes, P., and Zhang,
- Q.: Chemical processing of water-soluble species and formation of secondary organic
- aerosol in fogs, Atmospheric Environment, 200, 158-166,
- https://doi.org/10.1016/j.atmosenv.2018.11.062, 2019.
- Kuang, Y., He, Y., Xu, W., Yuan, B., Zhang, G., Ma, Z., Wu, C., Wang, C., Wang, S.,
- Zhang, S., Tao, J., Ma, N., Su, H., Cheng, Y., Shao, M., and Sun, Y.: Photochemical

Aqueous-Phase Reactions Induce Rapid Daytime Formation of Oxygenated Organic 451 Aerosol on the North China Plain, Environ. Sci. Technol., 2020. 452 Lim, Y. B., Tan, Y., Perri, M. J., Seitzinger, S. P., and Turpin, B. J.: Aqueous chemistry 453 and its role in secondary organic aerosol (SOA) formation, Atmospheric Chemistry and 454 Physics, 10, 10521–10539, https://doi.org/10.5194/acp-10-10521-2010, 2010. 455 Liu, H., Liu, T., Li, Y., Ge, A., Wang, L., Lai, S., Niu, G., Yin, J., Zhou, X., Liu, Y., 456 Wang, J., Zha, Q., Qi, X., Nie, W., Chi, X., Lou, S., Huang, X., Zhang, Y., Song, W., 457 Wang, X., and Ding, A.: Impacts of heatwaves on characteristics of atmospheric 458 methylglyoxal in a suburban area in eastern China, Journal of Geophysical Research: 459 Atmospheres, 130, e2025JD044284, https://doi.org/10.1029/2025JD044284, 2025. 460 Liu, R., Liu, T., Huang, X., Ren, C., Wang, L., Niu, G., Yu, C., Zhang, Y., Wang, J., Qi, 461 X., Nie, W., Chi, X., and Ding, A.: Characteristics and sources of atmospheric ammonia 462 at the SORPES station in the western yangtze river delta of China, Atmospheric 463 Environment, 318, 120234, https://doi.org/10.1016/j.atmosenv.2023.120234, 2024. 464 Liu, Z., Hu, B., Ji, D., Cheng, M., Gao, W., Shi, S., Xie, Y., Yang, S., Gao, M., Fu, H., 465 Chen, J., and Wang, Y.: Characteristics of fine particle explosive growth events in 466 beijing, China: Seasonal variation, chemical evolution pattern and formation 467 mechanism, Science The Total Environment, 687, 1073-1086, 468 https://doi.org/10.1016/j.scitotenv.2019.06.068, 2019. 469 McNeill, V. F.: Aqueous organic chemistry in the atmosphere: Sources and chemical 470 processing of organic aerosols, Environ. Sci. Technol., 49, 1237-1244, 471 https://doi.org/10.1021/es5043707, 2015.

- 472 Ng, N. L., Canagaratna, M. R., Jimenez, J. L., Chhabra, P. S., Seinfeld, J. H., and
- Worsnop, D. R.: Changes in organic aerosol composition with aging inferred from
- 474 aerosol mass spectra, Atmospheric Chemistry and Physics, 11, 6465-6474,
- https://doi.org/10.5194/acp-11-6465-2011, 2011.
- Paatero, P. and Tapper, U.: Positive matrix factorization: A non-negative factor model
- with optimal utilization of error estimates of data values, Environmetrics, 5, 111–126,
- https://doi.org/10.1002/env.3170050203, 1994.
- Peng, J., Hu, M., Shang, D., Wu, Z., Du, Z., Tan, T., Wang, Y., Zhang, F., and Zhang,
- R.: Explosive secondary aerosol formation during severe haze in the north China plain,
- Environ. Sci. Technol., 55, 2189–2207, https://doi.org/10.1021/acs.est.0c07204, 2021.
- Pye, H. O. T., Nenes, A., Alexander, B., Ault, A. P., Barth, M. C., Clegg, S. L., Collett
- Jr., J. L., Fahey, K. M., Hennigan, C. J., Herrmann, H., Kanakidou, M., Kelly, J. T., Ku,
- I.-T., McNeill, V. F., Riemer, N., Schaefer, T., Shi, G., Tilgner, A., Walker, J. T., Wang,
- 485 T., Weber, R., Xing, J., Zaveri, R. A., and Zuend, A.: The acidity of atmospheric
- particles and clouds, Atmospheric Chemistry and Physics, 20, 4809-4888,
- https://doi.org/10.5194/acp-20-4809-2020, 2020.
- Rogers, M. J., Joo, T., Hass-Mitchell, T., Canagaratna, M. R., Campuzano-Jost, P.,
- Sueper, D., Tran, M. N., Machesky, J. E., Roscioli, J. R., Jimenez, J. L., Krechmer, J.
- E., Lambe, A. T., Nault, B. A., and Gentner, D. R.: Humid summers promote urban
- aqueous-phase production of oxygenated organic aerosol in the northeastern united
- states, Geophysical Research Letters, 52, e2024GL112005,
- https://doi.org/10.1029/2024GL112005, 2025.

- Seinfeld, J. H. and Pankow, J. F.: Organic atmospheric particulate material, Annu. Rev.
- Phys. Chem., 54, 121–140,
- https://doi.org/10.1146/annurev.physchem.54.011002.103756, 2003.
- Sullivan, A. P., Hodas, N., Turpin, B. J., Skog, K., Keutsch, F. N., Gilardoni, S.,
- Paglione, M., Rinaldi, M., Decesari, S., Facchini, M. C., Poulain, L., Herrmann, H.,
- Wiedensohler, A., Nemitz, E., Twigg, M. M., and Collett Jr., J. L.: Evidence for ambient
- dark aqueous SOA formation in the Po Valley, Italy, Atmos. Chem. Phys., 16, 8095-
- 8108, https://doi.org/10.5194/acp-16-8095-2016, 2016.
- Sun, P., Nie, W., Chi, X., Xie, Y., Huang, X., Xu, Z., Qi, X., Xu, Z., Wang, L., Wang,
- T., Zhang, Q., and Ding, A.: Two years of online measurement of fine particulate nitrate
- in the western Yangtze River Delta: influences of thermodynamics and N<sub>2</sub>O<sub>5</sub> hydrolysis,
- Atmos. Chem. Phys., 18, 17177–17190, https://doi.org/10.5194/acp-18-17177-2018,
- 2018.
- Sun, W., Wang, D., Yao, L., Fu, H., Fu, Q., Wang, H., Li, Q., Wang, L., Yang, X., Xian,
- A., Wang, G., Xiao, H., and Chen, J.: Chemistry-triggered events of PM<sub>2.5</sub> explosive
- growth during late autumn and winter in shanghai, China, Environmental Pollution,
- 254, 112864, https://doi.org/10.1016/j.envpol.2019.07.032, 2019.
- Sun, Y., Du, W., Fu, P., Wang, Q., Li, J., Ge, X., Zhang, Q., Zhu, C., Ren, L., Xu, W.,
- Zhao, J., Han, T., Worsnop, D. R., and Wang, Z.: Primary and secondary aerosols in
- Beijing in winter: sources, variations and processes, Atmospheric Chemistry and
- Physics, 16, 8309–8329, https://doi.org/10.5194/acp-16-8309-2016, 2016.
- Sun, Y. L., Zhang, Q., Anastasio, C., and Sun, J.: Insights into secondary organic aerosol

- formed via aqueous-phase reactions of phenolic compounds based on high resolution
- mass spectrometry, Atmospheric Chemistry and Physics, 10, 4809-4822,
- https://doi.org/10.5194/acp-10-4809-2010, 2010.
- Tilgner, A., Schaefer, T., Alexander, B., Barth, M., Collett Jr., J. L., Fahey, K. M., Nenes,
- 520 A., Pye, H. O. T., Herrmann, H., and McNeill, V. F.: Acidity and the multiphase
- chemistry of atmospheric aqueous particles and clouds, Atmos. Chem. Phys., 21,
- 13483–13536, https://doi.org/10.5194/acp-21-13483-2021, 2021.
- Ulbrich, I. M., Canagaratna, M. R., Zhang, Q., Worsnop, D. R., and Jimenez, J. L.:
- Interpretation of organic components from Positive Matrix Factorization of aerosol
- mass spectrometric data, Atmos. Chem. Phys., 2009.
- Wang, F., Lv, S., Liu, X., Lei, Y., Wu, C., Chen, Y., Zhang, F., and Wang, G.:
- Investigation into the differences and relationships between gasSOA and aqSOA in
- winter haze pollution on Chongming Island, Shanghai, based on VOCs observation,
- Environmental Pollution, 316, 120684, https://doi.org/10.1016/j.envpol.2022.120684,
- 2023.
- Wang, J., Ge, X., Chen, Y., Shen, Y., Zhang, Q., Sun, Y., Xu, J., Ge, S., Yu, H., and
- Chen, M.: Highly time-resolved urban aerosol characteristics during springtime in
- yangtze river delta, China: Insights from soot particle aerosol mass spectrometry,
- Atmospheric Chemistry and Physics, 16, 9109–9127, https://doi.org/10.5194/acp-16-
- 9109-2016, 2016.
- Wang, J., Ye, J., Zhang, Q., Zhao, J., Wu, Y., Li, J., Liu, D., Li, W., Zhang, Y., Wu, C.,
- Xie, C., Qin, Y., Lei, Y., Huang, X., Guo, J., Liu, P., Fu, P., Li, Y., Lee, H. C., Choi, H.,

- Zhang, J., Liao, H., Chen, M., Sun, Y., Ge, X., Martin, S. T., and Jacob, D. J.: Aqueous
- production of secondary organic aerosol from fossil-fuel emissions in winter Beijing
- haze, Proc. Natl. Acad. Sci. U.S.A., 118, e2022179118,
- https://doi.org/10.1073/pnas.2022179118, 2021.
- Wexler, A. S. and Clegg, S. L.: Atmospheric aerosol models for systems including the
- ions H<sup>+</sup>, NH<sub>4</sub><sup>+</sup>, Na<sup>+</sup>, SO<sub>4</sub><sup>2-</sup>, NO<sub>3</sub><sup>-</sup>, Cl<sup>-</sup>, Br<sup>-</sup>, and H<sub>2</sub>O, Journal of Geophysical Research:
- Atmospheres, 107, ACH 14-1-ACH 14-14, https://doi.org/10.1029/2001JD000451,
- 2002.
- Wu, Y., Ge, X., Wang, J., Shen, Y., Ye, Z., Ge, S., Wu, Y., Yu, H., and Chen, M.:
- Responses of secondary aerosols to relative humidity and photochemical activities in
- an industrialized environment during late winter, Atmospheric Environment, 193, 66-
- 78, https://doi.org/10.1016/j.atmosenv.2018.09.008, 2018.
- Xian, J., Cui, S., Chen, X., Wang, J., Xiong, Y., Gu, C., Wang, Y., Zhang, Y., Li, H.,
- Wang, J., and Ge, X.: Online chemical characterization of atmospheric fine secondary
- aerosols and organic nitrates in summer Nanjing, China, Atmospheric Research, 290,
- 106783, https://doi.org/10.1016/j.atmosres.2023.106783, 2023.
- Xiao, Y., Hu, M., Li, X., Zong, T., Xu, N., Hu, S., Zeng, L., Chen, S., Song, Y., Guo, S.,
- and Wu, Z.: Aqueous secondary organic aerosol formation attributed to phenols from
- biomass burning, Science of The Total Environment, 847, 157582,
- https://doi.org/10.1016/j.scitotenv.2022.157582, 2022.
- Xu, W., Han, T., Du, W., Wang, Q., Chen, C., Zhao, J., Zhang, Y., Li, J., Fu, P., Wang,
- Z., Worsnop, D. R., and Sun, Y.: Effects of Aqueous-Phase and Photochemical

- Processing on Secondary Organic Aerosol Formation and Evolution in Beijing, China,
- Environ. Sci. Technol., 51, 762–770, https://doi.org/10.1021/acs.est.6b04498, 2017.
- Xu, W., Sun, Y., Wang, Q., Zhao, J., Wang, J., Ge, X., Xie, C., Zhou, W., Du, W., Li, J.,
- Fu, P., Wang, Z., Worsnop, D. R., and Coe, H.: Changes in Aerosol Chemistry From
- 2014 to 2016 in Winter in Beijing: Insights From High-Resolution Aerosol Mass
- Spectrometry, Journal of Geophysical Research: Atmospheres, 124, 1132–1147,
- https://doi.org/10.1029/2018JD029245, 2019.
- Yasmeen, F., Sauret, N., Gal, J.-F., Maria, P.-C., Massi, L., Maenhaut, W., and Claeys,
- 568 M.: Characterization of oligomers from methylglyoxal under dark conditions: A
- pathway to produce secondary organic aerosol through cloud processing during
- nighttime, Atmospheric Chemistry and Physics, 10, 3803–3812,
- https://doi.org/10.5194/acp-10-3803-2010, 2010.
- Zhao, J., Qiu, Y., Zhou, W., Xu, W., Wang, J., Zhang, Y., Li, L., Xie, C., Wang, Q., Du,
- W., Worsnop, D. R., Canagaratna, M. R., Zhou, L., Ge, X., Fu, P., Li, J., Wang, Z.,
- Donahue, N. M., and Sun, Y.: Organic Aerosol Processing During Winter Severe Haze
- Episodes in Beijing, JGR Atmospheres, 124, 10248–10263,
- https://doi.org/10.1029/2019JD030832, 2019.
- Zhou, J., Elser, M., Huang, R.-J., Krapf, M., Fröhlich, R., Bhattu, D., Stefenelli, G.,
- Zotter, P., Bruns, E. A., Pieber, S. M., Ni, H., Wang, Q., Wang, Y., Zhou, Y., Chen, C.,
- Xiao, M., Slowik, J. G., Brown, S., Cassagnes, L.-E., Daellenbach, K. R., Nussbaumer,
- T., Geiser, M., Prévôt, A. S. H., El-Haddad, I., Cao, J., Baltensperger, U., and Dommen,
- 581 J.: Predominance of secondary organic aerosol to particle-bound reactive oxygen

| 582 | species activity in fine ambient aerosol, Atmospheric Chemistry and Physics, 19, |         |            |       |      |       |     |            |
|-----|----------------------------------------------------------------------------------|---------|------------|-------|------|-------|-----|------------|
| 583 | 14703-14720, https://doi.org/10.5194/acp-19-14703-2019, 2019.                    |         |            |       |      |       |     |            |
| 584 | Ziemann, P. J. and Atkinson, R.: Kinetics, products, and mechanisms of secondary |         |            |       |      |       |     |            |
| 585 | organic                                                                          | aerosol | formation, | Chem. | Soc. | Rev., | 41, | 6582–6605, |
| 586 | https://doi.org/10.1039/C2CS35122F, 2012.                                        |         |            |       |      |       |     |            |
| 587 |                                                                                  |         |            |       |      |       |     |            |