# Peer review of "Characteristics and processing of aqueous secondary organic aerosols"

_EGUsphere, 2025_

## Referee Comment (RC1)

This manuscript presents a comprehensive characterization of aqueous-phase secondary organic aerosol (aqSOA) formation in suburban Nanjing during the wintertime of 2020, based on high-resolution AMS measurements, PMF source apportionment, and the estimated ALW and aerosol acidity. The study identifies two distinct diurnal patterns of aqSOA driven respectively by aqueous-phase chemistry and daytime photochemical oxidation. The findings provide meaningful insights into the evolving roles of nitrate, ALW, and acidity in SOA formation mechanisms in Eastern China. Overall, the manuscript is clearly written, scientifically sound, and suitable for publication after dealing with the below comments:

**1. Line 21. The statement that "its formation mechanisms remain unclear due to limited observational evidence" is somewhat overstated. A more precise phrasing would be: "...its formation pathways under real ambient conditions in Chinese urban regions remain insufficiently constrained..." This better reflects the current state of knowledge without implying a lack of global understanding. Same for Line 46-47.**

**2. Line 92. Replace "more" with "abundant"**

**3. In Section 2.3 (Line 123), the authors calculate aerosol liquid water (ALW) solely based on inorganic aerosol thermodynamics using E-AIM. However, recent field and laboratory studies have demonstrated that organic aerosol (OA) also contributes substantially to ALW, especially under high-RH conditions that are relevant in this study. For typical urban environments, OA hygroscopicity ($\kappa \approx 0.08–0.20$) can lead to organic-associated ALW that is comparable in magnitude to inorganic-associated ALW, particularly when OA mass concentrations are high (Kuang et al., 2021; Nguyen et al., 2016; Zhang et al., 2024).**

Given that one of the central conclusions of the manuscript is that ALW strongly modulates aqSOA formation, it is important to either: (1) Include OA-derived ALW in the analysis (e.g., using κ-Köhler parameterizations with measured OA composition), or (2) Provide a justification for omitting it—for example, demonstrating that inorganic ALW dominates under the specific composition and RH conditions at this site, or showing that including OA-ALW would not change the interpretation of Figures 2–3. At minimum, a short discussion acknowledging the potential magnitude of OA-ALW and its implications for the aqSOA–ALW relationship should be added.

*References:*
*Kuang, Y., et al. (2021). Contrasting effects of secondary organic aerosol formations on organic aerosol hygroscopicity. ACP, 21, 10375–10391.*
*Nguyen, T. K. V., et al. (2016). Liquid water: ubiquitous contributor to aerosol mass. ES&T Letters, 3, 257–263.*

*Zhang, J., et al. (2024). Quantified organic aerosol subsaturated hygroscopicity by a simple optical system. ACP, 24, 13445–13456.*

**4. Lines 147–153 should be moved to a position before Line 138. As currently written, the discussion at Line 138 is confusing because the manuscript begins interpreting aqSOA characteristics before explaining how aqSOA is identified.**

**5. :Line 197-203,the authors reported a strong correlation between aqSOA and total nitrate and interpreted this as evidence that nitrate enhances aqSOA formation. However, it is possible that organic nitrate ($NO_{3,org}$) contributes significantly to the total nitrate signal in the AMS, particularly under humid conditions, and that the apparent correlation may partly reflect joint formation pathways involving biogenic VOCs (e.g., isoprene, monoterpenes, etc.). Numerous studies (Boyd et al., 2015, 2017; Takeuchi & Ng, 2019; Zhang et al., 2020) have shown that aqueous and multiphase processing of organic nitrates can produce highly oxygenated organics with a tight relationship between aqSOA and $NO_{3,org}$.**

I would suggest the author separates $NO_3$ to inorganic and organic compounds following the method described by Farmer et al. (2010), and analyze how each component ($NO_{3,inorg}$ vs. $NO_{3,org}$) relates to aqSOA. This would largely benefit Section 3.2 discussion.

*Reference:*
*Boyd, C. M., et al. (2017). Secondary organic aerosol (SOA) from nitrate radical oxidation of monoterpenes: Effects of temperature, dilution, and humidity on aerosol formation, mixing, and evaporation. Environmental Science & Technology, 51(14), 7831–7841.*
*Boyd, C. M., et al. (2015). Secondary organic aerosol formation from the β-pinene+NO3 system: Effect of humidity and peroxy radical fate. Atmospheric Chemistry and Physics, 15, 7497–7522.*
*Takeuchi, M., & Ng, N. L. (2019). Chemical composition and hydrolysis of organic nitrate aerosol formed from hydroxyl and nitrate radical oxidation of α-pinene and β-pinene. Atmospheric Chemistry and Physics, 19, 12,749–12,766.*
*Zhang, J., et al. (2020). Evolution of aerosol under moist and fog conditions in a rural forest environment: Insights from high-resolution aerosol mass spectrometry. Geophysical Research Letters, 47(19), p.e2020GL089714.*
*Farmer, D. K., et al. (2010). Response of the aerosol mass spectrometer to organonitrates and organosulfates and implications for field studies. PNAS 2010, 107(15), 6670–6675.*